# The Forest Stakeholders' Perception towards the NATURA 2000 Network in the Czech Republic

**Jiří Schneider [1], Aleš Ruda [1,*] , Žaneta Kalasová [1] and Alessandro Paletto [2]**

[1] Department of Environmentalistics and Natural Resources, Faculty of Regional Development and International Studies, tř. Gen. Píky 7, Brno 61300, Czechia; jiri.schneider@mendelu.cz (J.S.); zaneta.kalasova@mendelu.cz (Ž.K.)

[2] Research Centre for Forestry and Wood, Council for Agricultural Research and Economics (CREA), p.za Nicolini, 6-38123 Trento, Italy; alessandro.paletto@crea.gov.it

* Correspondence: ruda@mendelu.cz; Tel.: +420-545-136-252

**Abstract:** Natura 2000 is a network of European protected areas, established under the provision of two directives of the European Union: the Habitats Directive (92/43/EEC) and the Birds Directive (79/409/EEC; 2009/147/EU). The Natura 2000 network can be considered an interesting instrument to maintain and improve ecosystem services provided by protected sites. The European Union member countries are free to organize the participatory process in the implementation of the Natura 2000 network. The participatory process is often overlooked despite it being an important tool to increase the social acceptance and reduce conflicts among stakeholders with different interests. The aim of the present study is to investigate the stakeholders' perceptions towards the ecosystem services provided by the Natura 2000 sites in the Czech Republic. The data was collected through a questionnaire survey involving 53 stakeholders (forester managers and nature conservation authorities) in all regions of the Czech Republic. The results show that for the respondents, the implementation of the Habitats and Birds Directives in the Czech Republic is very or quite important (54.7%), but at the same time, many respondents consider the Natura 2000 network an obstacle for economic activities close to the sites (66.0% of total respondents). In accordance with the stakeholders' opinions, the three most important human activities near and inside the Natura 2000 sites are agricultural activities, followed by nature conservation interventions and forestry activities. The representatives of environmental Non-Governmental Organizations (NGOs) and academia emphasize the importance of nature conservation interventions, while the other groups of interest consider the provisioning services supplied by agricultural and forestry activities as the most relevant ecosystem services. The results of this study can be considered as the starting point aimed to improve the participatory process in the establishment and management of the Natura 2000 sites based on the stakeholders' feelings and opinions.

**Keywords:** nature conservation; Natura 2000 forest sites; ecosystem services; stakeholders' involvement; questionnaire survey

## 1. Introduction

In recent decades, there has been an increasing interest in the socio-economic assessment of the ecosystem services provided by protected areas [1,2]. This renewed interest is due to the key role of protected areas, e.g., national and regional parks, biosphere reserves and Natura 2000 sites, for biodiversity conservation at a landscape level associated with the provision of additional ecosystem services such as tourist-attracting landscapes, environmental education opportunities and agricultural and forestry products [3]. As emphasized by Maes et al. [4], biodiversity conservation in the protected

areas also has the potential to maintain or improve the supply of ecosystem services. Therefore, protected areas can be considered of primary importance not only to mitigate biodiversity loss [5], but also to maintain and improve the other three categories of ecosystem services recognized by the Millennium Ecosystem Services (2005): provisioning, regulating and cultural services. The provisioning services include food, fodder, timber and water, the regulating services comprise water and carbon cycle regulation and natural hazards protection, while the cultural services consider recreational, spiritual, religious and other non-material benefits [6,7]. In the protected areas, the habitats and species biodiversity conservation—included in the category of supporting services—can be considered the main purpose [8].

At the European level, biodiversity conservation is pursued through the establishment of the Natura 2000 network under the Habitats Directive (92/43/EEC) and the Birds Directive (79/409/EEC; 2009/147/EU). The Natura 2000 is a network of protected areas covering Europe's most valuable and threatened species and habitats—117 million ha corresponding to 17% of the surface area of the EU countries—aimed to conserve nature diversity in Europe, taking into account the economic, social and cultural requirements at the national level [9,10]. The approach suggested by the European Commission to implement the Natura 2000 network in the EU member countries must be based on combining human activities and nature conservation aims [11]. The implementation of the Natura 2000 network can be addressed, both at the national and local level, through different approaches and support from local, regional and national authorities of state administration and self-government. In the implementation of the Natura 2000 network, the EU member countries are free to organize the participatory process for the management of these sites. According to Pezdevšek Malovrh et al. [3], in many EU member countries, the implementation of the Natura 2000 network is a controversial top-down process where public actors have the ultimate decision-making power, while the other stakeholders are consulted or informed. However, the participatory process is an important tool to increase the social acceptance related to the constraints imposed by the conservative measures of the Natura 2000 sites and to reduce the conflicts among groups of interest [11]. According to Humphreys [12], the long-term success of the Natura 2000 network must be based on the collaboration among civil society organizations, private and public actors in order to take in consideration the needs and interests of all the players involved. As shown by some studies, private actors, e.g., farmers and forest owners, consider provisioning services (agricultural and forestry products) as the most important ecosystem services also inside or near protected areas, while for environmental NGOs and public authorities, the most important categories are supporting (nature conservation) and cultural (recreational activities and environmental education) services, respectively [8,13–15].

The issue of the stakeholders' involvement in the designation of the Natura 2000 sites in the EU member countries has been addressed by Bouwma et al. [16,17]. Then, Nastran and Pirnat [18] showed the stakeholder involvement in the designation of the Natura 2000 sites in Slovenia. Those authors emphasized that the stakeholders' interest in the decision-making related to nature conservation is growing in recent decades. In particular, landowners, protected areas administrators and the general public have shown great interest in their active involvement in the decision-making process. However, in some countries, the participatory process aimed to involve stakeholders in the implementation of the Natura 2000 network is characterized by a low level of inclusiveness and transparency. In these cases, the participatory process is based on a top-down approach focused on involving national and local public administrations rather than representatives of civil society.

In the scientific literature, Apostolopoulou et al. [19] carried out an analysis of the participatory process in the management of the Greek Natura 2000 sites, while Rojas-Briales [20] investigated the key socio-economic issues in the implementation of the Natura 2000 network in Spain. In both those studies, the role and weight of local stakeholders, e.g., municipalities and landowners, is emphasized as an "ingredient" for the public acceptance of nature conservation policies. Chmielewski and Głogowska [21] focused on the conflicts related to information and communication technologies in randomly selected Natura 2000 sites in Poland. Those authors showed that from the perspective of local stakeholders,

the Natura 2000 sites are an opportunity for sustainable regional development. On the other hand, the above-mentioned study showed that the intensive development of Poland's infrastructure causes numerous conflicts in the case of investments in renewable energy resources (e.g., wind farms, biogas plants). Besides, Pietrzyk-Kaszyńska et al. [22] highlighted that local people's opposition towards protected areas is the result of restrictions imposed on landowners and local communities and a perceived unequal distribution of costs and benefits between social actors.

Starting from these considerations, the aim of this study is to analyze the stakeholders' perception towards the implementation of the Natura 2000 network in the Czech Republic. The idea of this investigation is to highlight the strengths and weaknesses of the participatory process and the perceived importance of ecosystem services provided by the Natura 2000 sites according to the stakeholders' opinions. The study was carried out within the Internal Grant Agency of the Faculty of Forestry at Mendel University within the project: Stakeholder Engagement in the Natura 2000 Network Implementation in the Czech Republic. This study conducted in the Czech Republic is part of the COST Action FP1207 "Orchestrating forest related policy analysis in Europe" (ORCHESTRA) aimed at investigating the participatory process in the implementation of the Natura 2000 network in some EU member countries (i.e., Czech Republic, Italy, Slovakia, Slovenia).

## 2. Materials and Methods

### 2.1. Natura 2000 Network in Czech Republic

During the European Union (EU) pre-accession phase, a comprehensive field mapping of natural habitats [23] took place in the Czech Republic (period: 2001–2004). The results of this survey have been used in the Habitat Catalogue of the Czech Republic and to designate the protected sites of the Natura 2000 network [24]. Afterwards, the Natura 2000 network was adopted in the Act No. 114/1992 Coll. on Nature and Landscape Protection. The Ministry of the Environment of the Czech Republic prepared a list of proposed Sites of Community Importance (SCIs) and Special Protection areas (SPAs) with the coordination of the Agency for Nature Conservation and Landscape Protection of the Czech Republic (ANCLP CR). Then, the list of SCIs was designated in the Government Order, while each SPA was designated in the individual Government Order. The final list (Government Order 132/2005 Coll.) formed by 863 areas located in the Continental and Pannonic regions (Government Order 132/2005 Coll.) was adopted by the European Commission.

Regarding the participatory process related to the implementation of the Natura 2000 network, the Czech Society of Ornithology coordinated the process by involving specialists based on individual personal contracts. Concurrently, some environmental NGOs developed a parallel priority list of sites with a high biodiversity value [25].

In 2016, the official list of sites designated for the selected species and habitat types was expanded in accordance with the conclusions of the bilateral meeting between the Czech Republic and the European Commission, which was held in March 2011 in Prague.

Currently, the Natura 2000 network in the Czech Republic includes 1153 sites—1112 SCIs and 41 SPAs—covering 111,500 hectares (14% of the total national land area) [26]. In the Natura 2000 network in the Czech Republic, there are 60 types of biotopes (19 priority habitats), 39 plant species (15 priority habitats) and 65 animal species (8 priority species), according to the Habitats Directive. In addition, there are 59 bird species included in the Birds Directive.

In the Habitat Catalogue of the Czech Republic, a total of 156 natural habitats and 19 non-natural habitats are included. In the rarest natural habitats of the Natura 2000 network, there are vegetation of annual halophilous grasses (M2.4), low xeric shrubs, secondary vegetation with *Prunus tenella* (K4B), river gravel banks with *Myricaria germanica* (M4.2), macrophyte vegetation of naturally eutrophic and mesotrophic still waters (with *Aldrovanda vesiculosa*) (V1E), cliff vegetation in the Sudeten cirques (A5), calcareous fens with *Cladium mariscus* (M1.8) and a *Salix lapponum* subalpine scrub (A8.1). All these natural habitats are characterized by a low number of sites and/or a small area; therefore, they are in

the category of critically endangered habitats in the Czech Republic. In the non-natural habitats, there are some production forests such as forest plantations of allochtonous coniferous trees (2022.37 km$^2$ in the Natura 2000 sites) and forest plantations of allochtonous deciduous trees (61.05 km$^2$) [24].

The typical forest habitats in the Czech Republic include the following (from lowlands to mountains): acidophilous oak forests, thermophilous oak forests, hardwood forests of lowland rivers, oak-hornbeam forests, beech forests or montane *Calamagrostis* spruce forests.

The typical land use of the Natura 2000 localities are forests, pastures or extensively managed meadows (with mowing), fresh water areas, wetlands and also, but rarely, old extensive orchards.

## 2.2. Research Framework

To assess the stakeholders' perceptions towards the implementation of the Natura 2000 network in the Czech Republic, the study was divided into four main steps.

The first step focuses on the stances and possible views of stakeholders during the implementation of Natura 2000 according to some key variables: the relative importance of human activities in the Natura 2000 sites, opportunities for and obstacles to human activities in the Natura 2000 sites, engagement of stakeholders in the decision-making process of the Natura 2000 network, democracy and cooperation versus conflicts between stakeholders.

In the second step, the stakeholders were identified and classified considering the delimitation of sites (stakeholder analysis). The main criteria used were the size and diversity of habitats, and the site specifics in relation to conflicts between nature protection and the interests of the stakeholders (e.g., buildings with bat colonies in European areas of conservation). The study is targeted at all regions of the Czech Republic and at the following eight groups of interest: municipality administration and forest managers, regional authorities, municipalities, water managers, interest groups, environmental non-governmental organizations, specially protected areas administration and churches. This paper presents the partial results of the pilot questionnaire survey and focuses on the attitudes of the three most-represented groups of respondents—foresters (municipality administration with forestry knowledge and professional forest managers), representatives of nature conservation authorities (NCA) and members of environmental NGOs. These are complemented by the "other" group, which includes academia and research institutions, landscape engineers (envi-planners) and the local municipality administration (i.e., municipalities). The questionnaires were distributed throughout the Czech Republic. Firstly, for reasons of representativeness, secondly, because of the need to consider the different approaches in different regions.

In the third step, a semi-structured questionnaire was developed and administered to the sample of stakeholders. The final version of the questionnaire (Appendix A)—which was developed after the pre-test stage—is formed by 25 closed-ended and open-ended questions divided into three thematic sections. The first thematic section focuses on the personal information of respondents including the number of years of expertise in the forestry or nature conservation sector. The second thematic section focuses on the personal perspectives of the participants on the relationship between nature conservation and human activities emphasizing the importance of ecosystem services provided by the Natura 2000 sites. The third thematic section focuses on the respondents assessed in the participatory process in the implementation of Natura 2000 (e.g., level of inclusiveness, transparency of the process, participatory techniques, conflicts among stakeholders and level of trust in other stakeholders). The data were collected between April 2017 and June 2018 by email administration. The collected data were aggregated by groups of interest (i.e., nature conservation authorities, forest managers, environmental NGOs, envi-planners, academia and research institutes, and municipality administration) and cross-compared.

## 2.3. Data Processing

The data collected with the questionnaire have been processed to highlight the perception of stakeholders, distinguishing between the forest managers and representatives of nature conservation

authorities. The answers to the open-ended questions were examined to find logical interpretations through a textual analysis, while the answers to the closed-ended questions were used to provide the main descriptive statistics (frequency distribution, mean, median, standard deviation). For the statistical data processing, the XLStat 2017 software by Addinsoft (Paris, FR) was used.

Regarding the question about the relationship between nature conservation and human activities (Section 2), the respondents assessed, through a pairwise comparison approach, the importance of five human activities related to the ecosystem services provision (Table 1).

**Table 1.** Human activities in the Natura 2000 sites and effects on ecosystem services provision.

| Human Activity | Definition | Ecosystem Services |
|---|---|---|
| Nature conservation interventions | All practices aimed to preserve and improve the natural environment and biodiversity | Maintenance and improvement of habitats and species biodiversity (supporting services) |
| Agricultural activities | Activities achievable in the Natura 2000 sites in accordance with the restrictions established by the current legislation | Increases in the agricultural products supply (provisioning services) |
| Forestry activities | Activities aimed to improve the productive function (timber and bioenergy production) of forests in accordance with the restrictions contained in the current legislation | Increases in the forest products supply such as timber, fuelwood, and woodchips (provisioning services) |
| Recreational activities | Non-consumptive recreational activities such as hiking, bird watching, wildlife viewing and relaxing | Improvement of recreational attractiveness of site (cultural services) |
| Environmental education activities | Activities aimed to increase people's knowledge and awareness about the environment and associated challenges | Increasing citizens' awareness and the cultural values of forests (cultural services) |

The data of the comparison between alternative human activities were processed using the analytic hierarchy process (AHP) method. The AHP is a multiple-criteria decision-making method used to rank the alternatives by taking into account the importance of the different criteria in order to facilitate decision-making for complex choices [27–29]. In the present study, the AHP method was applied with the aim of defining an order of priority for human activities in the Natura 2000 sites. The priority score of each human activity was used as an indicator of stakeholders' individual preference for activities and related ecosystem services. The consistency of the respondents' information is confirmed by the consistency ratio (*CR*). The value of the *CR* should be lower or equal to 0.1, otherwise preferences are not expressed consistently.

## 3. Results

Eighty-seven stakeholders were contacted by email explaining the objective of the study and asking them for willingness to participate in the survey. Fifty-three stakeholders were willing to participate in the survey (response rate of 61%), and thus were divided into six groups of interest (Table 2): nature conservation authorities (24.5% of respondents), forest managers (35.9% of respondents), environmental NGOs (18.9%), academia and research institutes (7.5%), municipality administration (7.5%) and envi-planners (5.7%).

**Table 2.** List of stakeholders involved in the survey.

| Group of Interest | Name of Stakeholder | Number of Respondents |
|---|---|---|
| Nature Conservation authorities | Regional Office | 5 |
| | Protected Landscape Area Administration | 3 |
| | National Park Administration | 2 |
| | Czech Environmental Inspectorate | 1 |
| | Ministry of Environment | 2 |
| Forest managers | Forests CR, s.p. State Enterprise | 5 |
| | Military Forests and Farms, s.p. | 5 |
| | Municipal and town forests | 6 |
| | Others | 3 |
| Environmental NGOs | | 10 |
| Academia and research institutes | | 4 |
| Envi-planners | | 3 |
| Municipality administration | | 4 |
| | Total | 53 |

Regarding the geographical distribution of the respondents, the representatives of all the main regions of the Czech Republic are involved in the survey (Figure 1).

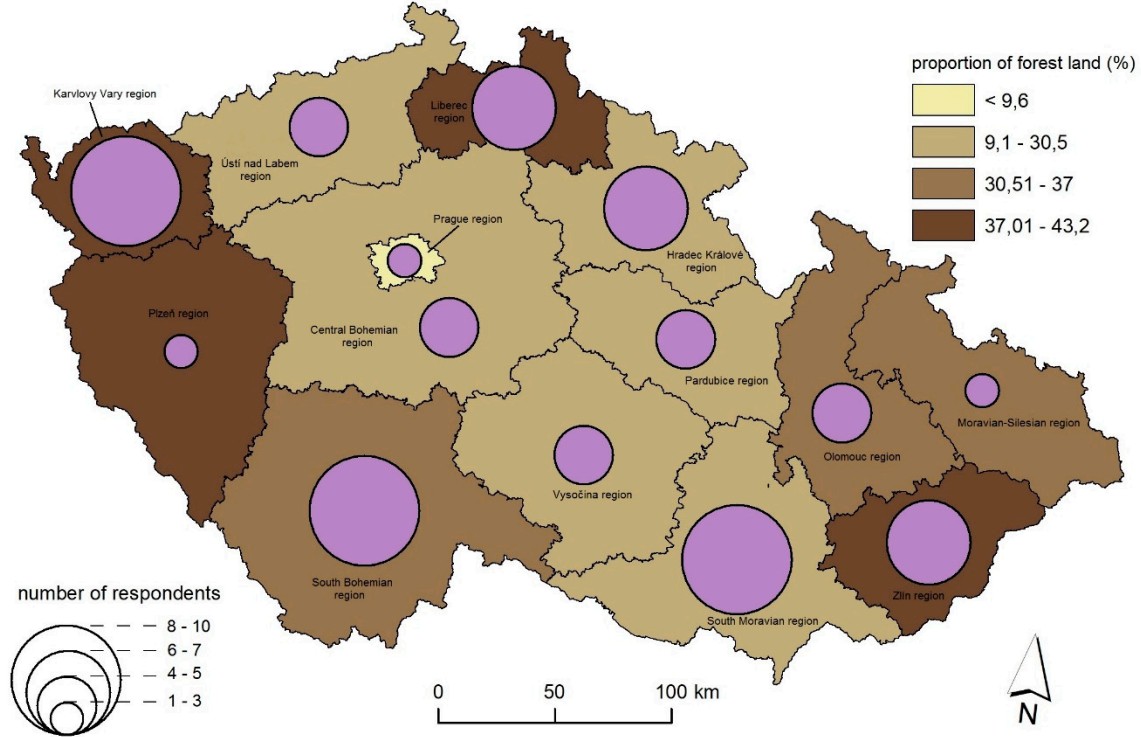

**Figure 1.** Number of respondents by Czech Republic region.

The respondents have a medium-high level of expertise, with an average working time in their organization/association of 11.8 years (11.9 years for the representatives of nature conservation authorities; 12.8 years for forest managers; 11.4 years for environmental NGOs; 7.0 years for envi-planners; 5.5 years for municipality administration; and 12.3 years for academics). The distribution in classes can be summarized as follows: 41.6% of respondents have worked in their organization/association for less than 10 years, 26.4% for 10 to 14 years, 15.1% for 15 to 19 years and the remaining 17.0% for more than 19 years.

The results of Section 2 show that the respondents consider the implementation of the Habitats and Birds Directives in the Czech Republic to be very or quite important (54.7% of respondents), while only 9.4% of respondents consider it little important and 5.4% not important.

The majority of respondents (72% and 66%, respectively) consider the Natura 2000 network as an opportunity or an obstacle for human activities with an economic outcome within and around the Natura 2000 sites.

The main obstacles highlighted by the respondents can be summarized as follows: increasing bureaucracy, including conflicts with nature conservation authorities, and the request of different than preferred management approaches (e.g., longer rotation period, request on higher woody debris, higher portion of autochthonous tree species, protection of endangered species).

The main opportunities emphasized by the respondents are the following: attractiveness of localities for tourists, subsidies for nature conservation management and a lower level of management intensity.

The results of the pairwise comparison show that for stakeholders ($n = 53$), the most important human activities in the Natura 2000 network is agricultural activities (priority score $w = 0.2540$), followed by nature conservation interventions ($w = 0.2463$) and forestry activities ($w = 0.2153$). Conversely, for the stakeholders, the recreational activities and environmental education activities have a marginal importance with a priority score of 0.1479 and 0.1365, respectively. Therefore, the Czech stakeholders

emphasize the importance of the provisioning services (agricultural and forestry products) also near and inside the Natura 2000 sites.

Observing the results by groups of interest, some interesting differences are highlighted (Table 3): for the representatives of nature conservation authorities and envi-planners, the most important human activity is agricultural activity (priority score $w = 0.2657$ and $0.3070$, respectively), while for the representatives of environmental NGOs and academia, the most important human activity is nature conservation interventions (priority score $w = 0.3025$ and $0.2946$, respectively). As expected, for the representatives of forest managers and municipalities, the most important human activity near and inside the Natura 2000 sites is forestry activities with a priority score of $0.2758$ and $0.2828$, respectively. Considering the relationship between human activities and ecosystem services, the results show that the different groups of interest have different needs, perceptions and preferences towards ecosystem services provision in the protected areas: provisioning services (food, timber and wood biomass for bioenergy) are considered priority for four out of the four groups of interest (nature conservation and forest managers, envi-planners and municipality administration), while supporting services are considered priority for the other two groups of interest (environmental NGOs and academia).

**Table 3.** Priority scores for the human activities in the Natura 2000 sites by group of interest.

| Activity/Group | Nature Conservation Authorities ($n = 13$) | Forest Managers ($n = 18$) | Environmental NGOs ($n = 10$) | Envi-planners ($n = 3$) | Municipality Administrations ($n = 4$) | Academia and Research Institutes ($n = 4$) |
|---|---|---|---|---|---|---|
| Nature conservation interventions | 0.2592 | 0.2057 | **0.3025** | 0.2838 | 0.1790 | **0.2946** |
| Agricultural activities | **0.2657** | 0.2293 | 0.2567 | **0.3070** | 0.2460 | 0.2563 |
| Forestry activities | 0.2017 | **0.2758** | 0.1582 | 0.1483 | **0.2828** | 0.1610 |
| Recreational activities | 0.1441 | 0.1362 | 0.1415 | 0.1491 | 0.1940 | 0.1658 |
| Environmental education activities | 0.1292 | 0.1529 | 0.1411 | 0.1118 | 0.098 | 0.1223 |

**Note**: values in bold: the most important human activity for each group of interest

In Section 3, the respondents provided their opinion on the appropriate level of inclusiveness in the implementation of the Natura 2000 network. According to the principles of the direct citizen participation approach, all citizens should be involved in the decision-making process, while for the interest group participation approach, only organized groups (i.e., public administrations, associations, private organizations) should be involved in the decision-making process. The results show that for 52.4% of respondents, only organized groups should be involved in the decision-making process, while for the remaining 47.6% of respondents, both organized groups and citizens should be involved (Table 4). By observing the data by group of interest (Table 4), the results show that the majority of nature conservation authorities and environmental NGOs positively consider the involvement of citizens in the decision-making process (63.6% and 62.5%, respectively), while for the forest managers, it would be more appropriate to involve only organized groups (73.3%).

**Table 4.** Appropriate level of inclusiveness of organized groups and citizens in the implementation of the Natura 2000 network.

| Type of Social Actors/Group | Nature Conservation Authorities ($n = 13$) | Forest Managers ($n = 18$) | Environmental NGOs ($n = 10$) | Envi-planners ($n = 3$) | Municipality Administrations ($n = 4$) | Academia and Research Institutes ($n = 4$) |
|---|---|---|---|---|---|---|
| Organized groups | 36.4% | 73.3% | 37.5% | 50.0% | 50.0% | 52.4% |
| Organized groups and citizens | 63.6% | 26.7% | 62.5% | 50.0% | 50.0% | 47.6% |

The results show that 79% of the respondents were involved in the participatory process for the implementation of the Natura 2000 network in at least one of the three levels (national, regional and local level). In particular, 60.4% of stakeholders have been involved in the participatory process at the national level, 52.8% at the regional level and only 15.1% at the local level. These results show a gap in

the involvement of stakeholders at the local level compared with the national level. By observing the data by group of interest, it is interesting to emphasize that some groups—forest managers (47.8% at the national, 39.1% at the regional and 13.0% at the local level) and municipality administrations (100% at the national level)—have been involved mainly at the national level, while other groups—nature conservation authorities (40.0% at the national, 55.0% at the regional and 5.0% at the local level), academia and research institutes (25.0% at the national, 50.0% at the regional and 25.0% at the local level)—have been involved mainly at the regional level. Regarding the phase of the implementation process, 95.0% of respondents confirmed their involvement during the design phase, while 37.0% of respondents were involved in the other two phases (management and monitoring and the evaluation phase). Concerning the last two phases, the main groups of interest involved in the participatory process were (Figure 2) nature conservation authorities (36.4% in the management phase and 22.7% in the monitoring phase) and envi-planners (25.0% in both phases).

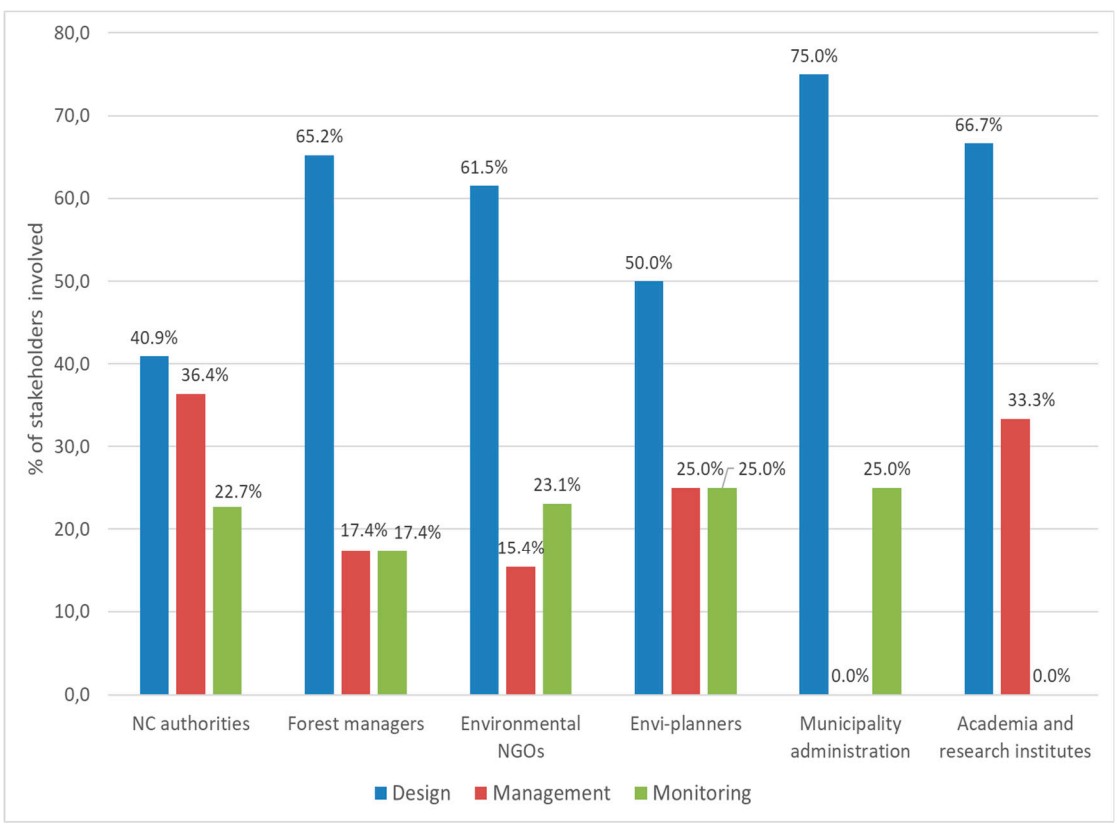

**Figure 2.** Stakeholders involved (%) in the implementation phases of the Natura 2000 network by group of interest.

Considering participatory techniques used in the implementation process, the respondents were mostly involved in public meetings (60% of total respondents), working groups (62%) and focus groups (33%). Other participatory techniques such as brainstorming (16%), facilitation (2%), scenario techniques (2%) and online forums (5%) were used less frequently.

The results show that 55% of respondents were satisfied with the participation process adopted in the implementation of the Natura 2000 network. According to the stakeholders' opinions, the two main strengths of the participatory process were the opportunity to express their opinions (95% of satisfied respondents) and the inclusion of local knowledge in the decision-making process (76%). Conversely, 45% of respondents were not satisfied with the participatory process mainly because decisions were made without taking into account their suggestions and comments (81% of unsatisfied respondents).

Considering the level of involvement of other stakeholders (information, consultation, collaboration and co-decision) in the implementation of the Natura 2000 network, it can be stated

that the European Union, Ministry of Environment, Ministry of Agriculture, regions and provinces participated mostly at all levels of involvement. Academia and research institutes, environmental NGOs and municipality administrations have been involved at the information, consultation and collaboration level. Only farmers and hunting associations were mainly informed. According to the respondents' opinions, the participatory process in the implementation of the Natura 2000 network did not change the level of trust in the other stakeholders.

Considering the two most important groups of interest (nature conservation and forest managers, Table 5), the responses reflect the fact that the workers of the addressed nature conservation authorities have been dealing with the implementation of the Natura 2000 network for a long time. Their work is not only to discuss all levels of the implementation with the other stakeholders but specially to deal with the final implementation, i.e., the declaration of the protected area and the elaboration of the management principles. Consequently, they perceive the context comprehensively with regard to the objective of the implementation.

**Table 5.** Approach of the representatives of nature conservation authorities and forest managers in the implementation of the Natura 2000 network in the Czech Republic.

| Issue | Evaluation by Nature Conservation Authorities | Evaluation by Forest Managers |
|---|---|---|
| Importance of Natura 2000 implementation at national level in CR | Predominantly positive evaluation, one neutral opinion, no negative opinions | Ambiguous attitude, variability of opinions (on the scale from very important to completely useless) |
| Natura 2000 as an opportunity for activities carried out for the purpose of economic result. Reasoning | Natura 2000 is also an opportunity to implement activities carried out for the purpose of economic profit. One dissenting opinion. The mechanism is mainly indirect– increasing the landscape attractiveness for recreational activities and soft tourism. Grant is another option. | Slightly prevailing opinion that Natura 2000 can also represent an opportunity for economic activities, mainly tourism and subsidies as a mechanism |
| Natura 2000 as a barrier to activities carried out for the purpose of economic result. Reasoning | Opinions are divided in half. The main reason for restricting economic activities are the limits resulting from nature protection. | A clear obstacle (only two people do not perceive the system as an obstacle). The main mechanism are the restrictions on farming |
| Perception of relations between Environmental protection—Agricultural activities—Recreational activities—Environmental education—Forest productive function | There is no unambiguous stance. Respondents mostly perceive agricultural activities as more important than nature conservation (!), recreational activities and environmental education equally important as forest productive function | No unequivocal or significantly prevailing opinion on the superiority of its significance over the others |
| Did you have an opportunity to express your opinion in the process of negotiating and approving the Natura 2000 network implementation? | All respondents were involved in the negotiations. | Two thirds of the respondents could express their opinion on the process, one third of the respondents could not. |
| Does everybody have an equal opportunity to express their opinion in the process of negotiating and approving the Natura 2000 network implementation? | Everybody has an opportunity to express their opinion on the Natura 2000 network implementation. | Not all stakeholders have the same opportunity to express their opinion on the network implementation |
| Were your opinions and comments considered? | All respondents' views and comments were considered | The narrow majority of the respondents stated that their opinions and comments were not considered |
| Were your comments satisfactorily respected? | Yes | The narrow majority of the respondents were not satisfied with respect given to their comments |
| What should the level of stakeholder involvement in the process be like? | Accentuating the role of the Ministry of the Environment, cooperation between NGOs and owners is important, the Ministry of Agriculture should only act as an advisor | Most significant at the Ministry of the Environment and the Ministry of Agriculture |
| The level of your trust in stakeholders before and after the experience with Natura 2000 implementation | Most of the experience of the implementation process did not affect the level of trust in individual stakeholders. Generally, the Ministry of the Environment and universities and research institutions enjoy the highest level of trust, whereas hunting associations enjoy the lowest. Regions have a better picture of the situation after the completion of the process | Experience of the implementation process did not affect the trust in individual stakeholders. It is low for the European Union and the Ministry of the Environment, medium for the Ministry of Agriculture and regions. Forest owners are the most trustworthy |
| Conflicts arising during the implementation of Natura 2000. What were they about? | All respondents encountered conflicts. The principle lies primarily in mistrust of nature conservation authorities and insufficient awareness of the owners in the process. | Practically all respondents (except two) perceived conflicts during the implementation. The main reason was the non-acceptance of the comments of foresters and forest owners by the nature conservation authorities |
| Satisfaction with the outcome of the processes in which the respondents were involved | The respondents are satisfied with the outcome of the processes they were involved in | With only one exception, none of the respondents were satisfied with the outcome of the implementation processes they were involved in |

The forest managers' responses reflect their responsibility for the property, workers and management in general, the need to generate profit and the different internal approaches to forest ecosystems and forest management.

The intensive requirements for a profitable forest management focusing on its wood-producing function naturally conflict with the principles of nature conservation and ecological stability. This is

the reason why many of them cannot be met, which leads to dissatisfaction with the acceptance of the foresters' comments.

At the same time, the responses reflect the experience of how the requirements for the different records of documents and official procedures complicate and stretch out the administrative management.

The fundamental difference between this and the first group is that the Natura 2000 implementation is the main objective of work of the addressed officials, whereas for the foresters, it is just one of (the less important) aspects that they have to deal with.

## 4. Discussion and Conclusions

The main objective of the survey—developed in four EU member countries (Czech Republic, Italy, Slovakia, and Slovenia)—was to investigate the strengths and weaknesses of the participatory process in the implementation of the Natura 2000 network. In the survey implemented in the Czech Republic, the level of importance of human activities in the Natura 2000 sites was used as an indicator of the stakeholders' perceptions and preferences on ecosystem services.

The results of present study can be compared to studies conducted in the other three countries: Italy [29], Slovakia [10] and Slovenia [14]. In the Italian context, De Meo et al. [30] showed that the most important activity in the Natura 2000 sites is nature conservation interventions, followed by environmental education and agricultural activities. Likewise, Gallo et al. [14] highlighted a Slovenian stakeholder preference for nature conservation interventions, forest activities and agricultural activities. In the Slovakian context, Brescancin et al. [10] pointed out that for stakeholders, the three most important activities are nature conservation interventions, environmental education and agricultural activities. Conversely, in the present study, nature conservation interventions are not considered the most important activity, but the Czech stakeholders emphasize more the importance of agricultural and forestry activities related to the provisioning services supply. This aspect is in line with the integration approach of the EU which emphasizes the combination of human activities (e.g., recreational activities, agricultural and forestry practices) and nature conservation purposes in the Natura 2000 sites. As emphasized by Pechanec et al. [24], the majority of the habitat types in the Czech Republic require various levels of human interventions or extensive farming to maintain a stable habitat character. This is due to the characteristics and location of many non-forest habitat types in the Czech Republic [24,31]. Therefore, the coexistence between human activities (agricultural and forestry activities) and nature conservation measures in the Natura 2000 sites is a key point in the Czech Republic, as mentioned by the stakeholders involved in our survey.

The highly perceived importance of provisioning services in the Czech Republic is related to the key role of the forestry sector in the national economy. In 2016, forestry production was 13,827 thousand m$^3$ of industrial roundwood and 2,336 m$^3$ of fuelwood, corresponding to a gross value added (GVA) of €883 million and a 14,800 FTE (full-time equivalent) employment [32]. In addition, it is interesting to highlight that the majority of forests in the Czech Republic are production forests (approximately 75% of the total forest area), followed by special purpose forests (22%) and protection forests (3%) [33]. Therefore, the stakeholders involved in this survey emphasized the need for the coexistence between nature conservation and forestry activities inside and close to the Natura 2000 sites. The coexistence between these two human activities refers to the trade-off between the supporting services (e.g., maintenance of habitats and species diversity) and provisioning services (e.g., industrial roundwood and fuelwood production). Particularly, an intensive forest management, i.e., clearcuttings and the removal of wood residues and stumps for energy production, can have negative impacts on forest biodiversity [34,35] and protection against natural hazards [36]. However, an extensive forest management based on a close-to-nature approach can significantly reduce the negative effects of industrial roundwood and bioenergy production both on the supporting and regulating services [37].

Regarding the importance of the provisioning services in the EU member countries, Tsiafouli et al. [9] highlighted the wide spread of human activities, e.g., agricultural and forestry,

hunting, fishing, urbanization, transportation and tourism, in the Natura 2000 sites (86% of sites are affected by human activities). Those authors asserted that nature conservation initiatives could succeed only by combining socio-economic and ecological sustainability. The results of that study are in line with the opinions of the Czech stakeholders involved in our study.

A common finding among the above-mentioned studies is an intensive clash between nature conservation and forest management, which has been long-term and stems primarily from the uncompromising attitudes of both parties. Concerning the obstacles for human activities in and around the boundaries of the Natura 2000 sites, many stakeholders highlighted the constraints of productive forest management practices. In addition, a distrust and negative experience within almost all groups of interest in the participatory process related to the implementation processes are evident. As expected, a fundamental finding is the clear consensus among all groups of interest that the implementation process is full of clashes and conflicts, mainly due to the completely different attitudes of the stakeholders. From the view of forest managers and environmental NGOs, this is also a consequence of the lack in the communication process by the authorities.

The results of the present study show the role of the participatory process in the establishment and management of the Natura 2000 sites. In the Czech Republic, as in the other EU countries involved in the survey (Italy, Slovenia, Slovakia), the need for a greater involvement of stakeholders in the decision-making process related to nature conservation is considered a key point that needs to be improved. In accordance with the Aarhus Convention on Access to Information, Public Participation in Decision-Making and Access to Justice in Environmental Matters (1998), the public authority should encourage transparent information and an open and effective public participation during the environmental decision-making. Unfortunately, the implementation of the Natura 2000 network was a controversial top-down process in which the stakeholders were involved differently from country to country (from information to collaboration/co-decision). Similarly, also in the management of the Natura 2000 sites, the groups of interest are involved in a different way. In order to increase the social acceptance and reduce conflicts in the establishment and management of the Natura 2000 sites, a standard protocol concerning the public participation should be adopted in all EU member countries. All groups of interest with different interests, needs and expectations must be included in the participatory process with special regard to those stakeholders affected by the constraints related to nature conservation measures. In this context, the consultation of stakeholders, as in the present survey in the Czech Republic, can be considered as the starting point of a participatory process able to include the opinions, needs and requests of different groups of interest. In addition, the consultation of stakeholders can provide useful information on the perceived importance of ecosystem services supporting national and local decision makers.

The results of this survey show a unanimous agreement that respondents across the groups of interest perceive the Natura 2000 network as a barrier to economic activities, mainly due to the restrictions arising from the nature conservation requirements. At the same time, it is mostly perceived as an opportunity for the development of some economic activities. The main mechanism is tourism and recreational activities associated with maintaining the attractiveness of the protected area. Another support mechanism is the drawing of subsidies for close-to-nature farming, although the amount is insufficient in many cases as pointed out by representatives of forest managers and municipality administration. The respondents only partially disagree on the evaluation of the significance of the Natura 2000 network in the Czech Republic. Employees of the Department of Nature Conservation, members of environmental NGOs and representatives of the "other" groups of interest agree on its significance, while the representatives of forest managers do not have a unanimous view on this issue.

It is worth noticing that in terms of the importance of the selected activities, the Department of Nature Conservation's workers perceive agricultural activities as more important than all the others, including nature conservation and environmental education. On the other hand, environmental NGOs and the "other" groups of interest agree on the importance of nature conservation.

The future steps of the study will investigate some key aspects related to the maintenance and improvement of ecosystem services in the Natura 2000 sites based on stakeholders' opinions and point of views. The data collected in the four EU member countries will be used to define a standardized protocol for public participation in the Natura 2000 sites.

**Author Contributions:** Conceptualization, J.S. and A.R.; methodology, J.S.; software, A.R.; validation, J.S., Ž.K. and A.P.; formal analysis, A.R..; investigation, Ž.K.; resources, J.S.; data curation, Ž.K.; writing—original draft preparation, J.S and Ž.K.; writing—review and editing, J.S,, A.R., Z.K. and A.P.; visualization, A.R.; supervision, J.S. and A.P.; project administration, J.S.; funding acquisition, J.S. All authors have read and agreed to the published version of the manuscript.

**Funding:** This research received no external funding.

**Acknowledgments:** Questionnaire was developed under the COST Action "ORCHESTRA". Own interview survey was made in relation to the project EHP-CZ02-OV-1-032-2015: Raising awareness and publicity of the importance of forest functions in the landscape and near-natural watercourses in urban areas as a part of basin ecosystem services.

**Conflicts of Interest:** The authors declare no conflict of interest.

## Appendix A

We are conducting a survey about the implementation of European Union (EU) Habitats Directive and Birds Directive in Czech Republic with a special emphasis on participatory processes and stakeholders' involvement in the decision-making process.

We are asking your help, as expert or stakeholder, for understanding the reality of participatory processes in the implementation (***Implementation****: consists of a wide range of actions and decisions that include multiple interpretations and applications of policies by various people*) of Natura 2000 network in Czech Republic at local level. Hoping to have your support, we are sending you these questions.

Completely filled-in questionnaires will be essential for our research.

Please, answer as a representative of your organization/association.

**Section 1—Personal information**

1.1. Name of organization/association:_______________________________________________

1.2. Role in the organization/association:_____________________________________________

1.3. Years of work in your organization/association:____________________________________

**Section 2—Natura 2000 network perceptions**

2.1. Do you know which percentage of land area is covered by Natura 2000 sites in your Czech region?

□ less than 10% □ 10–20% □ 20–30% □ 30–40% □ more than 40% □ no opinion

2.2. In your opinion the Habitats and Birds Directives implementation in Czech Republic is important for the nature conservation at national level?

□ very important □ quite important □ averagely important □ little important □ not important

2.3. In your opinion the Natura 2000 network is an **opportunity** for human activities with an economic outcome, direct or indirect, in and around the boundaries of the protected sites?

□ YES □ NO

If YES, could you explain the reasons:

_____________________________________________________________________________________

_____________________________________________________________________________________

2.4. In your opinion the Natura 2000 network is an **obstacle** for human activities with an economic outcome, direct or indirect, in and around the boundaries of the protected sites?

□ YES □ NO

If YES, could you explain the reasons:

_______________________________________________________________________________

_______________________________________________________________________________

2.5.　In your opinion how the following stakeholders should be involved in the Natura 2000 sites management decision making in Czech Republic?

| | Information | Consultation | Collaboration | Co-decision |
|---|---|---|---|---|
| Directly affected individuals (e.g., landowners) | ☐ | ☐ | ☐ | ☐ |
| Interested stakeholders (e.g., environmental NGOs, tourism associations, public administrations/managers) | ☐ | ☐ | ☐ | ☐ |
| General public (citizens) | ☐ | ☐ | ☐ | ☐ |

*Information: the level of participation which provides the public with balanced and objective information to assist them in understanding the problem, alternatives, opportunities and/or solutions (e.g., fact sheets, web sites, etc.).*

*Consultation: the level of participation which obtains public feedback on analysis, alternatives and/or decisions (e.g., focus group, surveys, public meetings).*

*Collaboration: the level of participation which engages the knowledge and resources of stakeholders (i.e., site-based events).*

*Co-decision: the level of participation which shares power and responsibility for the decision being made and their outcomes creating management groups.*

2.6.　Could you compare the importance given to nature conservation (biodiversity), agricultural activities, productive forest functions, recreational activities, environmental education and research activities in the Czech Natura 2000 network?

| | | | | | | |
|---|---|---|---|---|---|---|
| Nature conservation | ++ | + | equal | + | ++ | Agricultural activities |
| Recreational activities | ++ | + | equal | + | ++ | Nature conservation |
| Recreational activities | ++ | + | equal | + | ++ | Agricultural activities |
| Environmental education | ++ | + | equal | + | ++ | Nature conservation |
| Environmental education | ++ | + | equal | + | ++ | Recreational activities |
| Environmental education | ++ | + | equal | + | ++ | Agricultural activities |
| Productive forest functions | ++ | + | equal | + | ++ | Nature conservation |
| Productive forest functions | ++ | + | equal | + | ++ | Recreational activities |
| Productive forest functions | ++ | + | equal | + | ++ | Environmental education |
| Productive forest functions | ++ | + | equal | + | ++ | Agricultural activities |

**Section 3—Public participation in the implementation of Natura 2000**

3.1.　Have you been involved in the Natura 2000 implementation decision process?

☐ YES ☐ NO

If YES, describe/explain what was the role of the institution in which you are employed in this process and what was your personal role?

_______________________________________________________________________________

_______________________________________________________________________________

**Fill in the following part of the questionnaire if you have answered YES at the Question 3.1**

3.2.  Could you select the geographical level at which you have been involved?

□ Local level □ Regional level □ National level

3.3.  Could you select in which phase you have been involved?

□ Design □ Management □ Monitoring and Evaluation

*Design: phase during which the list of sites, to include in Natura 2000 network and to protect on the basis of the presence of habitats and species, is discussed and negotiated.*
*Management of the Natura 2000 sites at local level.*
*Monitoring and evaluation of the implementation process of the Habitats Directive and the assessment of its impacts on nature conservation.*

3.4.  For how long have you been involved in the implementation of Natura 2000 network?
_______________________________________________________________________________

3.5.  Which types of actors have been involved in the **implementation of Natura 2000 process** during your participation in it? □ only representatives of organized groups (public institutions, associations and private organizations) □ representatives of organized groups and individuals (citizens)

3.6.  Which kind of methods have been employed to enhance participation in the processes you were involved in?

□ Public meeting (*Public meeting: is a forum, a public event for information and discussion about subjects' perceptions on Natura 2000 network. The goal is informing and getting informed.*)

□ Focus group (*Focus groups: is a group discussion designed to learn about subjects' perceptions on Natura 2000 network. Focus groups rely on the dynamics of group interaction to reveal participants' similarities and differences of opinion.*)

□ Brainstorming (*Brainstorming: is a common method used in groups to help members think of as many ideas as possible. The members are encouraged to produce ideas as quickly as possible without considering the value of the idea. The emphasis is on quantity, not quality.*)

□ Working group (*Working group: small groups of experts aimed to discuss on specific thematic issues.*)

□ Mediation/Facilitation techniques (*Mediation/Facilitation techniques: are techniques in which the facilitator/mediator is a neutral third party, who ensures that the procedures are followed, and helps the participants to step out from their individual views and to define a common goal together.*)

□ Scenario techniques (*Scenario techniques: are techniques that use various scenarios enabling stakeholders to address a variety of issues across different geographies and at different scales.*)

□ On line forums (*On line forums: is a method aimed to collect views and information about Natura 2000 implementation.*)

□ Other_______________________________________

Were you satisfied with this approach(es)/method(s)?

□ very satisfied □ quite satisfied □ averagely satisfied □ not very satisfied □ not at all satisfied

3.7.  Did you have the opportunity to express your opinion in the process?

□ YES □ NO

3.8.  Do you think that all participants in process have an equal opportunity to express their views?

□ YES □ NO

If NO, could you explain the reasons:

_______________________________________________________________________________

_______________________________________________________________________________

3.9.　Do you think that your suggestions and comments in process have been taken into account?
　　　　　　　　　　　　　　　　□ YES □ NO

　　If YES were you satisfied with the feedback information consequently given to you?
　　　　　　□ very satisfied □ quite satisfied □ averagely satisfied □ not very satisfied □ not at all satisfied

3.10.　Which approach has been adopted to take decisions in the above mentioned processes?

　　□ Decision by authority (***Decision by authority***: *one person decides. The decision is taken by the most expert person or by a person who decides after listening to the group.*)

　　□ Minority decision (***Minority decision***: *small number of group member decides.*)

　　□ Democratic (majority) decision (***Democratic (majority) decision***: *everyone votes and the majority wins.*)

　　□ Consensus decision (***Consensus decision***: *everyone supports the solution even if not the favourite.*)

　　□ Unanimous decision (***Unanimous decision***: *everyone has to agree on a given solution/proposition.*)

　　□ No opinion/I don't know

3.11.　According to your opinion, the whole Natura 2000 implementation process has been **transparent**:

| 1 (not at all) | 2 | 3 | 4 | 5 (completely) |
|---|---|---|---|---|

3.12.　During the Natura 2000 implementation process, has the **local knowledge** been included?

　　□ NO

　　□ YES, please indicate in which way: _______________________________________________

　　_________________________________________________________________________________

3.13.　Have you been one of the organizers of the national communication campaign about Natura 2000 implementation?
　　　　　　　　　　　　　　　　□ YES □ NO

　　Which communication tools were used during the national communication campaign?

　　□ Mass media (e.g., local newspaper, radio, television)

　　□ Formal invitation

　　□ Social networks

　　□ Newsletters

　　□ Press releases

　　□ Others ___________________

3.14. Which **level of stakeholders' involvement** has been adopted in the Natura 2000 implementation process?

| | Information | Consultation | Collaboration | Co-decision |
|---|---|---|---|---|
| European Union | ☐ | ☐ | ☐ | ☐ |
| Ministry of Environment Land and Sea Protection | ☐ | ☐ | ☐ | ☐ |
| Ministry of Agricultural, Food and Forestry Policies | ☐ | ☐ | ☐ | ☐ |
| Regions and provinces | ☐ | ☐ | ☐ | ☐ |
| Universities and Research Centers | ☐ | ☐ | ☐ | ☐ |
| Environmental NGOs | ☐ | ☐ | ☐ | ☐ |
| Forest owners | ☐ | ☐ | ☐ | ☐ |
| Farms owners | ☐ | ☐ | ☐ | ☐ |
| Hunting associations | ☐ | ☐ | ☐ | ☐ |
| Municipalities | ☐ | ☐ | ☐ | ☐ |
| Others ___________________ | ☐ | ☐ | ☐ | ☐ |

3.15. Which is your **level of trust** regarding your relation with the stakeholders before and after (or during) the Natura 2000 implementation process?

| | **Before** Natura 2000 | | | **After** or **During** Natura 2000 | | |
|---|---|---|---|---|---|---|
| European Union | ☐ Low | ☐ Medium | ☐ High | ☐ Low | ☐ Medium | ☐ High |
| Ministry of Environment Land and Sea Protection | ☐ Low | ☐ Medium | ☐ High | ☐ Low | ☐ Medium | ☐ High |
| Ministry of Agricultural, Food and Forestry Policies | ☐ Low | ☐ Medium | ☐ High | ☐ Low | ☐ Medium | ☐ High |
| Regions and provinces | ☐ Low | ☐ Medium | ☐ High | ☐ Low | ☐ Medium | ☐ High |
| Universities and Research Centers | ☐ Low | ☐ Medium | ☐ High | ☐ Low | ☐ Medium | ☐ High |
| Environmental NGOs | ☐ Low | ☐ Medium | ☐ High | ☐ Low | ☐ Medium | ☐ High |
| Forest owners | ☐ Low | ☐ Medium | ☐ High | ☐ Low | ☐ Medium | ☐ High |
| Farms owners | ☐ Low | ☐ Medium | ☐ High | ☐ Low | ☐ Medium | ☐ High |
| Hunting associations | ☐ Low | ☐ Medium | ☐ High | ☐ Low | ☐ Medium | ☐ High |
| Municipalities | ☐ Low | ☐ Medium | ☐ High | ☐ Low | ☐ Medium | ☐ High |
| Others_________________ | ☐ Low | ☐ Medium | ☐ High | ☐ Low | ☐ Medium | ☐ High |

3.16. During the Natura 2000 implementation process have you noticed conflicts among stakeholders?

☐ YES ☐ NO

If YES, could you describe **which kind** of conflict and **actors involved**?

_______________________________________________________________________________

_______________________________________________________________________________

3.17. Are you in general satisfied with the results of the participatory processes you were involved in? (Choose between the following 5 and **explain why**)

☐ not at all _________________________________________________________________

☐ slightly satisfied __________________________________________________________

☐ no opinion________________________________________________________________

☐ highly satisfied____________________________________________________________

☐ extremely satisfied_________________________________________________________

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
