# Peer review of "The Forest Stakeholders’ Perception towards the NATURA 2000 Network in the Czech Republic"

_forests, doi:10.3390/f11050491_

Round 1

Reviewer 1 Report

Natura 2000 sites in the study area need to be better described. By assessing the prevalent land use to understand how much provisioning services affected compared to other functions.
Land use could influence stakeholders' comments due to the interests related to the production activities present.
To evaluate forest provisioning services, it is also necessary to know the type of forest management in progress. How much forest is managed and how.
The differences between the study "De Meo, I., Brescancin, F., Graziani, A., Paletto, A. Management of Natura 2000 sites in Italy: an exploratory study on stakeholder opinions" must be evaluated on the differences of the sites in the respective countries. The Italian Natura 2000 network consists mainly of forests and grasslands, in the Czech Republic it is not known.

Author Response

Responses are enclosed in an attachment.

Reviewer 2 Report

Abstract:.

After reading the abstract, I had the impression that the survey includes stakeholders with different interests (line 21), but among the respondents there were no representatives of agriculture (according the section 2.2). Also the results presents in Abstract suggest the huge role of agriculture, thus surprising is lack of representatives of agriculture sector. The Authors should reconsider this and put more emphasis on foresters (the title, the abstract, keywords doesn’t suggest in any way that the article concerns only forest sector representatives thus the  sampling may be called into question).

Introduction:

Lines 53-56 and 76-80 - it looks like a repeat of the same information. I'd leave it in the text only lines 76-80

Lines 84-87 – requires some corrections, reediting. Authors mention existence of some gap in participatory process, but they are not defining these gaps. Then, in next line they put information about “great interest among public” but they beginning the sentences as explanation of “gap”. The design of these sentences makes it difficult for me as a reader to determine what is this “gap”.

Line 88 – “In the European literature” - I don't understand why this emphasis in “European literature”? 

Line 98 – “Always for the Polish context” -- The statement "always" is very categorical. All the more so because Pietrzyk-KaszyĹ„ska did not do research for the whole Poland, but for mountainous areas, and the information to which the authors of the manuscript refer also refers to the works of authors from other countries. Thus I suggest elimination this “always”

Line 106-107 – This look rather as a part of Acknowledgment. After reading the whole text, it turned out that the Czech Republic is one of the 4-5 countries where the study was conducted. The authors neatly omitted this in the introduction, but referred to it in the discussion. Perhaps it is worthwhile to write something more about the fact that this is part of a larger whole, all the more so with the same survey being published in Annex of the manuscript (De Meo et al. 2016)

Materials and Methods

Line 161-162: - it looks like a repeat of the same information (line 159-160). Writing that the first thematic section concerns personal information adds nothing new to what is written in

Line 159-160. I suggest you to change this whole paragraph. For example, after the sentences: “The final version of questionnaire (Annex 1) – developed after the pre-test stage – is formed by 25 closed-ended and open-ended questions divided in three thematic sections” put a dot. And begin after that with the sentence form line 161: “The first thematic section focuses on the personal information of respondents”. And in consequences delete these information: “Personal information” (Section 1), “Natura 2000 network perceptions” (Section 2), and “Public participation in the implementation of Natura 2000” (Section 3)” , or redistribute them in better place.

Line 168-169 – why farmers are not included in study?

Table 1 and lines 182-194 could be connected, because now it seems like a repetition of the information. I suggest the preparation of table which includes 3 aspects: Human activity, Ecosystem services, and definition of human activities. 

Results

Line 208 – “At the end of the data collection” I'd remove that sentence, it looks like a huge randomness in the study. And suggest another questions like: i) how the process of contacting with respondent look like, ii) how many emails were sends iii) to whom emails were send, iv) what was the response rate. I suggest to put some information like: “The response rate to questionnaire was…with the … field questionnaire” if it is possible.

Line 212 – Is it really important information for the study? is it used in analysis somewhere? Figure 1 – is unnecessary, in Czech, poorly legible and not much to bring to work. If authors want such data, it may be better to combine it with Table 2, but I would rather remove it. And definitely I would recommend to remove Figure 1 in  such a look.

Line 214 – what does “medium-high expertise” mean? on what basis it was determined? And is this information included in materials and methods

Line 225 – ”the sample”  - remove

Lines 231-237  – Obstacles and opportunities – from which point of view it is presented, only from foresters’ or also for others? If only from foresters why only this group.

Line 237 – “obstacles and opportunities depend on individual experience of stakeholders” – how?? Where is it explained?

Line 238 – why n=52 not 53? Where is explanation?

Line 244-246  - it is a part of discussion not results.

Line 256-260 – the topic of ecosystem services was not the main issue in the questionnaire. The answers are like an indicator of ES perception and preferences, thus I think this should be part of discussion, because does not originate directly from the survey and was not the purpose of that survey (I suppose)

Lines 265- 271 – The results of section 3 – there is not enough information in materials and methods about this affected group and involvement. Maybe it could be more explained here. Not always people have access to Appendix (sometimes they read Appendix at the end). In first I thought that affected and interested stakeholders are distinguished from respondents.  Generally I would rewrite this part.

Line 277-279: “Regarding the participatory process for the implementation of Natura 2000 network, 79% of the respondents were involved in the participatory process for the implementation of Natura 2000 network in at least one of the three levels”  - I think there's a way to convey that information without writing the same thing twice.

Line 301-304, -   I think that sentences could be rewritten – but I am not native speaker.

Figure 3  and line 298-300 - do they relate to the same thing? This map fits better than Fig 1. given the information on it. It doesn't refer at all to what's written in the text. This is also in English not in Czech as at the Fig 1 – which I think can be plus.

Line 301-308 – English should be improved. It is difficult to read with understanding what is described here. I think this is because all of: “also, on the other hand, but” used in the text. It is difficult to follow the logic of the text

Table 5 and 6 – should be presented together as one table. It would be easier to read.

Discussion and conclusions

Line 350-365 – here are mentioned earlier articles based on the same questionnaire. If Authors put this information in Introduction, here they could only compare the results. Also priority score received by other scientist are unnecessary in this section, only general information here. When mentioning these other studies, the authors should try to justify these differences. Why only in Czech agriculture and forestry are more important than nature conservation? What the consequences might be?

Line 406-415 – The assessment of the survey from the methodological point of view, after 3-4 papers on this topic, and without mentioning this earlier as a purpose of this article is looks strange and should be reconsider by Authors.

Line 416-420 – This is probably the only place where the authors mention ecosystem services in the discussion, despite the fact that ES are set as one of the objectives of the study. This aspect has been described little and inaccurately. I would advise you to bend over to this topic. If production is a priority in the Czech Republic, other services are at risk and this is worth describing

Line 418-420 How will this study be implemented in other places ? and it has not been done already: Italy, Slovakia, Slovenia?

To sum up:

I like the Introduction part, the Results should be rewritten or English should be improved to make it more clear. This definitely will improve Discussion part. The Discussion needs more information about ecosystem services. I recommend to the Authors to rethink their conclusions from this work.

Author Response

(The authors gave the same response as above.)

Round 2

Reviewer 2 Report

I see overall improvement of the manuscript text and look. Hovewer I have still some remarks.

Lines 360-374, I cannot find the link between described results and questionnaire. Especially: "The results
show that for 52.4% of respondents only organized groups should be involved in the decision-making process, while for the remaining 47.6% of respondents both organized groups and citizens should be involved (Table 4)." but the question 3.6  is : Which types of actors have been involved? not should be involved. Maybe this consider something else, if yes please explain me this. 

Also the % are probably wrong: 52,4% and 47,6% concerns Academic and research in Table 4, and in text are presented as answers of all. 

361-364 this is explanation and should be rather put in introduction, or discussion part.: "According to the principles of the direct citizen participation approach all citizens should be involved in the decision-making process, while for the interest group participation approach only organized groups (i.e. public administrations, associations, private organizations) should be involved in the decision-making process.

Still there is lack of information why for example in Italy people prefer nature conservation and in Czech provisioning services. Maybe there is a difference in respondents group (in Czech – foresters in Italy – NGO, or environmentalists) – I don’t know?

Author Response

Comments are attached.
